# Sparsely-Connected Neural Networks: Towards Efficient VLSI Implementation of Deep Neural Networks

**Arash Ardakani, Carlo Condo and Warren J. Gross**
Department of Electrical and Computer Engineering
McGill University, Montréal, Québec, Canada
Email: arash.ardakani@mail.mcgill.ca, carlo.condo@mail.mcgill.ca, warren.gross@mcgill.ca

## ABSTRACT

Recently deep neural networks have received considerable attention due to their ability to extract and represent high-level abstractions in data sets. Deep neural networks such as fully-connected and convolutional neural networks have shown excellent performance on a wide range of recognition and classification tasks. However, their hardware implementations currently suffer from large silicon area and high power consumption due to the their high degree of complexity. The power/energy consumption of neural networks is dominated by memory accesses, the majority of which occur in fully-connected networks. In fact, they contain most of the deep neural network parameters. In this paper, we propose sparsely-connected networks, by showing that the number of connections in fully-connected networks can be reduced by up to 90% while improving the accuracy performance on three popular datasets (MNIST, CIFAR10 and SVHN). We then propose an efficient hardware architecture based on linear-feedback shift registers to reduce the memory requirements of the proposed sparsely-connected networks. The proposed architecture can save up to 90% of memory compared to the conventional implementations of fully-connected neural networks. Moreover, implementation results show up to 84% reduction in the energy consumption of a single neuron of the proposed sparsely-connected networks compared to a single neuron of fully-connected neural networks.

## 1 INTRODUCTION

Deep neural networks (DNNs) have shown remarkable performance in extracting and representing high-level abstractions in complex data (Lecun et al. (2015)). DNNs rely on multiple layers of interconnected neurons and parameters to solve complex tasks, such as image recognition and classification (Krizhevsky et al. (2012)). While they have been proven very effective in said tasks, their hardware implementations still suffer from high memory and power consumption, due to the complexity and size of their models. Therefore, research efforts have been conducted towards more efficient implementations of DNNs (Han et al. (2016)). In the past few years, the parallel nature of DNNs has led to the use of graphical processing units (GPUs) to execute neural networks tasks (Han et al. (2015)). However, their large latency and power consumption have pushed researchers towards application-specific integrated circuits (ASICs) for hardware implementations (Cavigelli et al. (2015)). For instance, in (Han et al. (2016)), it was shown that a DNN implemented with customized hardware can accelerate the classification task by $189\times$ and $13\times$, while saving $24{,}000\times$ and $3{,}400\times$ energy compared to CPU (Intel i7-5930k) and GPU (GeForce TITAN X), respectively.

Convolutional layers in DNNs are used to extract high level abstractions and features of data. In such layers, the connectivity between neurons follows a pattern inspired by the organization of the animal visual cortex. It was shown that the computation in the visual cortex can mathematically be described by a convolution operation (LeCun et al. (1989)). Therefore, each neuron is only connected to a few neurons based on a pattern and a set of weights is shared among all neurons. In contrast, in a fully-connected layer, each neuron is connected to every neuron in the previous and next layers and each connection is associated with a weight. These layers are usually used to learn non-linear

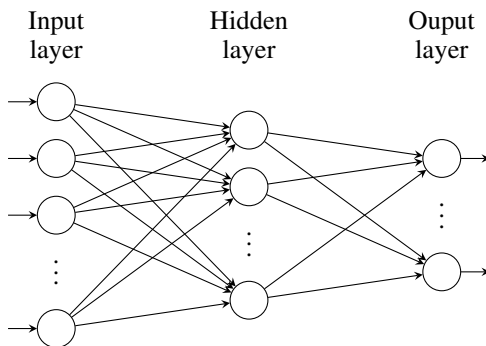

Figure 1: A two-layer fully-connected neural network

combinations of given data. Fig. 1 shows a two-layer fully-connected network. The main computation kernel performs numerous vector-matrix multiplications followed by non-linear functions in each layer. In (Courbariaux & Bengio (2016); Horowitz (2014); Han et al. (2016)), it was shown that the power/energy consumption of DNNs is dominated by memory accesses. Fully-connected layers, which are widely used in recurrent neural networks (RNNs) and adopted in many state-of-the-art neural network architectures (Krizhevsky et al. (2012); Simonyan & Zisserman (2014); Zeiler & Fergus (2013); Szegedy et al. (2015); Lecun et al. (1998)), independently or as a part of convolutional neural networks, contain most of the weights of a DNN. For instance, the first fully-connected layer of VGGNet (Simonyan & Zisserman (2014)), which is composed of 13 convolution layers and three fully-connected layers, contains 100M weights out of a total of 140M. Such large storage requirements in fully-connected networks result in copious power/energy consumption.

To overcome the aforementioned issue, a pruning technique was first introduced in (Han et al. (2015)) to reduce the memory required by DNN architectures for mobile applications. However, it makes use of an additional training stage, while information addresses identifying the pruned connections still need to be stored in a memory. More recently, several works have focused on the binarization and ternarization of the weights of DNNs (Courbariaux & Bengio (2016); Courbariaux et al. (2015); Lin et al. (2015); Kim & Smaragdis (2016)). While these approaches reduce weight quantization and thus the memory width, the number of weights is unchanged.

In (Shafiee et al. (2016b)), an alternative deep network connectivity named StochasticNet and inspired from the brain synaptic connection between neurons was explored on low-power CPUs. StochasticNet is formed by randomly removing up to 61% connections in both fully-connected and convolution layers of DNNs, speeding up the classification task.

In (Wen et al. (2016)), a method named structured sparsity learning (SSL) was introduced to regularize the convolutional layers' structures of DNNs. SSL can learn a structured sparsity of DNNs to efficiently speed up the convolutional computations both on CPU and GPU platforms.

In this paper, we propose sparsely-connected networks by randomly removing some of the connections in fully-connected networks. Random connection masks are generated by linear-feedback shift registers (LFSRs), which are also used in the VLSI implementation to disable the connections. Experimental results on three commonly used datasets show that the proposed networks can improve network accuracy while removing up to 90% of the connections. Additionally, we apply the proposed algorithm on top of the binarizing/ternarizing technique achieving a better misclassification rate than the best binarized/ternarized networks reported in literature. Finally, an efficient very large scale integration (VLSI) hardware architecture of a DNN based on sparsely-connected network is proposed, which saves up to 90% memory and 84% energy with respect to the traditional architectures.

The rest of the paper is organized as follows. Section 2 briefly introduces DNNs and their hardware implementation challenges, while Section 3 describes the proposed sparsely-connected network and their training algorithm. In Section 4 the experimental results over three datasets are presented and compared to the state of the art. Section 5 portrays the proposed VLSI architecture for the sparsely-connected network, and conclusions are drawn in Section 6.

## 2 PRELIMINARIES

### 2.1 DEEP NEURAL NETWORKS

DNNs are constructed using multiple layers of neurons between the input and output layers. These are usually referred to as hidden layers. They are used in many current image and speech applications to perform complex tasks as recognition or classification. DNNs are trained through an initial phase, called the learning stage, that uses data to prepare the DNN for the task that will follow in the inference stage. Two subcategories of DNNs which are widely used in detection and recognition tasks are convolutional neural networks (CNNs) and RNNs (Han et al. (2016)). Due to parameter reuse in convolutional layers, they are well-studied and can be efficiently implemented with customized hardware platforms (Chen et al. (2016); Shafiee et al. (2016a); Chen et al. (2016)). On the other hand, fully-connected layers, which are widely used in RNNs like long short-term memories and as a part of CNNs, require a large number of parameters to be stored in memories.

DNNs are mostly trained by the backpropagation algorithm in conjunction with stochastic gradient descent (SGD) optimization method (Rumelhart et al. (1986)). This algorithm computes the gradient of a cost function $C$ with respect to all the weights in all the layers. A common choice for the cost function is using the modified hinge loss introduced in (Tang (2013)). The obtained errors are then backward propagated through the layers to update the weights in an attempt to minimize the cost function. Instead of using a whole dataset to update parameters, data are first divided in mini-batches and parameters are updated using each mini-batch several times to speed up the convergence of the training algorithm. The weight updating speed is controlled by a learning rate $\eta$. Batch normalization is also commonly used to regularize each mini-batch of data (Ioffe & Szegedy (2015)): it speeds up the training process by allowing the use of a bigger $\eta$.

### 2.2 TOWARDS HARDWARE IMPLEMENTATION OF DNNS

DNNs have shown excellent performance in applications such as computer vision and speech recognition: since the number of neurons has a linear relationship with the ability of a DNN to perform tasks, high-performance DNNs are extremely complex in hardware. AlexNet (Krizhevsky et al. (2012)) and VGGNet (Simonyan & Zisserman (2014)) are two models comprising convolutional layers followed by some fully-connected layers, which are widely used in classification algorithms. Despite their very good classification performance, they require large amounts of memory to store the numerous parameters. Most of these parameters (more than 96%) lie in fully-connected layers. In (Han et al. (2016)), it was shown that the total energy of DNNs is dominated by the required memory accesses. Therefore, the majority of power in a DNN is dissipated through fully-connected layers of DNNs. Moreover, the huge memory requirements make possible only for very small DNNs to be fitted in on-chip RAMs in ASIC/FPGA platforms.

Recently, many works tried to reduce the computational complexity of DNNs. In (Akopyan et al. (2015)), the spiking neural network based on stochastic computing (Smithson et al. (2016)) was introduced, where 1-bit calculations are performed throughout the whole architecture. In (Ardakani et al. (2015)), integral stochastic computing was used to reduce the computation latency, showing that stochastic computing can consume less energy than conventional binary radix implementations. However, both works do not manage to reduce the DNN memory requirements.

Network pruning, compression and weight sharing have been proposed in (Han et al. (2016)), together with weight matrix sparsification and compression. However, additional indexes denoting the pruned connections are required to be stored along with the compressed weight matrices. In (Han et al. (2015)), it was shown that the number of indexes are almost the same as the number of non-zero elements of weight matrices, thus increasing the word length of the required memories. Moreover, the encoding and compression techniques require inverse computations to obtain decoded and decompressed weights, and introduce additional hardware complexity for hardware implementation compared to the conventional computational architectures. Other pruning techniques presented in literature such as (Anwar et al. (2015)) try to reduce the memory required to store the pruned locations by introducing a structured sparsity in DNNs. However, the resulting network yields up to 31.81% misclassification rate on the CIFAR-10 dataset.

---

**Algorithm 1:** Training algorithm for the proposed sparsely-connected network

---

**Data**: Fully-connected network with parameters $W$, $b$ and $M$ for each layer. Input data $x$, its corresponding targets $t$, and learning rate of $\eta$.

**Result**: $W$ and $b$

1 **1. Forward computations**

2 **for** *each layer i in range(1,N)* **do**

3 $W_s \leftarrow W_i \cdot M_i$

4 Compute layer output $y_i$ according to (3) and its previous layer output $y_{i-1}$, $W_s$ and $b_i$.

5 **end**

6 **2. Backward Computations**

7 Initialize output layers activation gradient $\dfrac{\partial C}{\partial y_N}$

8 **for** *each layer j in range(2,N-1)* **do**

9 Compute $\dfrac{\partial C}{\partial y_j}$

10 **end**

11 **for** *each layer j in range(1,N-1)* **do**

12 Compute $\dfrac{\partial C}{\partial W_s}$ knowing $\dfrac{\partial C}{\partial y_j}$ and $y_{j-1}$

13 Compute $\dfrac{\partial C}{\partial b_j}$

14 Update $W_j : W_j \leftarrow W_j - \eta \dfrac{\partial C}{\partial W_s}$

15 Update $b_j : b_j \leftarrow b_j - \eta \dfrac{\partial C}{\partial b_j}$

16 **end**

---

## 3 SPARSELY-CONNECTED NEURAL NETWORKS

Considering a fully-connected neural network layer with $n$ input and $m$ output nodes, the forward computations are performed as follow

$$y = act(Wx + b), \tag{1}$$

where $W$ represents the weights and $b$ the biases, while $act()$ is the non-linear activation function in which $\text{ReLU}(x) = max(0, x)$ is used in most cases (Nair & Hinton (2010)). The network's inputs and outputs are denoted by $x$ and $y$, respectively.

Let us introduce the sparse weight matrix $W_s$ as the element-wise multiplication

$$W_s = W \cdot M, \tag{2}$$

where $W_s$ and $M$ are sparser than $W$. The Mask binary matrix $M$ can be defined as

$$M_{n \times m} = \begin{bmatrix} M_{11} & M_{12} & \ldots & M_{1m} \\ M_{21} & M_{22} & \ldots & M_{2m} \\ \vdots & \vdots & \ddots & \vdots \\ M_{n1} & M_{n2} & \ldots & M_{nm} \end{bmatrix},$$

where each element of Mask $M_{ij} \in \{0,1\}$, $i \in \{1, \ldots, n\}$ and $j \in \{1, \ldots, m\}$. Note that the dimensions of $M$ are the same as the weight matrix $W$. Similarly to a fully-connected network (1), the forward computation of the sparsely-connected network can be expressed as

$$y = act(W_s x + b). \tag{3}$$

We propose the use of LFSRs to form each column of $M$, similar to the approach used in stochastic computing to generate a binary stream (Gaines (1969)). In general, an $nb$-bit LFSR serially generates $2^{nb} - 1$ numbers $S_i \in (0, 1), i \in \{1, 2, \ldots 2^{nb} - 1\}$. A random binary stream with expected value of $p \in [0\ 1]$ can be obtained by comparing $S_i$ with a constant value of $p$. This unit is hereafter referred

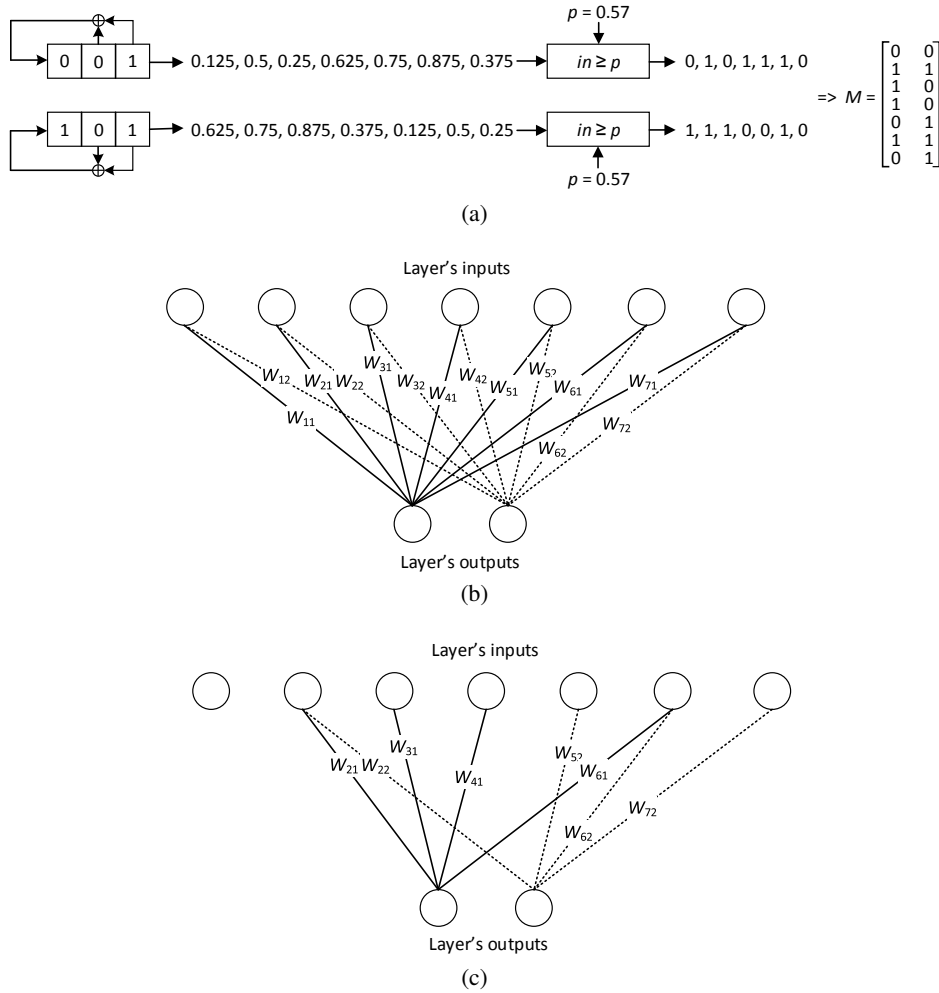

Figure 2: (a) shows the formation of a Mask matrix $M$ using a 3-bit LFSR for $p = 0.57$. (b) shows a fully-connected layer. (c) shows a sparsely-connected layer formed based on $M$.

to as stochastic number generator (SNG). Therefore, a random binary stream element $X_i \in \{0, 1\}$ is 1 when $S_i \geq p$, and 0 otherwise. Fig. 2 shows the formation of a small sparsely-connected network using binary streams generated by LFSR units. Fig. 2(a) shows a 3-bit LFSR unit with its 7 different values and a random binary stream with expected value of $p = 0.57$. A total of $m$ LFSRs of $\log_2(n)$-bit length with different seed values are required to form $M$. By tuning the value of $p$ it is possible to change the sparsity degree of $M$, and thus of the sparsely-connected network. Fig. 2(b) and Fig. 2(c) show the fully-connected network based on $W$ and the sparsely-connected version based on $W_s$.

Algorithm 1 summarizes the training algorithm for the proposed sparsely-connected network. The algorithm itself is very similar to what would be used with a fully-connected network, but considers each network layer to have a mask that disables some of the connections. The forward propagation (line 1-5) follows (3), while derivatives in the backward computations (line 6-16) are computed with respect to $W_s$. It is worth mentioning that most CNNs use fully-connected layers and the proposed training algorithm can still be used for those layers in CNNs.

## 4 EXPERIMENTAL RESULTS

We have validated the effectiveness of the proposed sparsely-connected network and its training algorithm on three datasets: MNIST (LeCun & Cortes (2010)), CIFAR10 (Krizhevsky (2009)) and SVHN (Netzer et al. (2011)) using the Theano library (Team (2016)) in Python.

Table 1: Misclassification rate for Different Network Sizes on MNIST

| Case | Method | Network Configuration | Misclassification Rate (%) | Number of Parameters |
|---|---|---|---|---|
| 1 | Fully-Connected | 784-512-512-10 | 1.18 | 669706 |
| | Sparsely-Connected 50% | 784-512-512-10 | 1.19 | 335370 |
| 2 | Fully-Connected | 784-256-256-10 | 1.35 | 269322 |
| | Sparsely-Connected 60% | 784-512-512-10 | 1.20 | 268503 |
| | Sparsely-Connected 70% | 784-512-512-10 | 1.31 | 201636 |
| 3 | Fully-Connected | 784-145-145-10 | 1.41 | 136455 |
| | Sparsely-Connected 80% | 784-512-512-10 | 1.28 | 134768 |
| 4 | Fully-Connected | 784-77-77-10 | 1.75 | 67231 |
| | Sparsely-Connected 90% | 784-512-512-10 | 1.75 | 67901 |
| 5 | Fully-Connected | 784-12-12-10 | 4.68 | 9706 |
| | Sparsely-Connected 90% | 784-100-100-10 | 3.16 | 8961 |

## 4.1 EXPERIMENTAL RESULTS ON MNIST

The MNIST dataset contains 60000 gray-scale $28 \times 28$ images (50000 for training and 10000 for testing), falling into 10 classes. A deep fully-connected neural network is used for evaluation and the hinge loss is considered as the cost function. The training set is divided into two separate parts. The first 40000 images are used as the training set and the rest for the validation and test sets. All models are trained using SGD without momentum, a batch size of 100, 500 epochs and the batch normalization method.

Table 1 summarizes the misclassification rate of sparsely-connected neural networks compared to fully-connected neural networks for different network configurations, using single-precision floating-point format. We adopted a fully-connected network with 784-512-512-10 network configuration as a reference network, in which each number represent the number of inputs to each fully-connected layer. From this, we formed sparse weight matrices $W_s$ with different sparsity degrees. For instance, sparsely-connected 90% denotes sparse weight matrices containing 90% zero elements. Case 1 shows that a sparsely-connected neural network with 50% fewer connections achieves approximately the same accuracy as the fully-connected network using the same network configuration. In Cases 2 and 3, the sparsely-connected networks with 60% and 80% fewer connections achieve a better misclassification rate than the fully-connected network while having approximately the same number of parameters. Case 4 shows no gain in performance and number of parameters for a sparsely-connected 90% and network configuration of 784-512-512-10 compared to the fully-connected at the same number of parameters. However, we can still reduce the connections up to 90% using a smaller network, as shown in Case 5.

Recently, BinaryConnect and TernaryConnect neural networks have outperformed the state-of-the-art on different datasets (Courbariaux et al. (2015); Lin et al. (2015)). In BinaryConnect, weights are represented with either -1 or 1, whereas they can be -1, 0 or 1 in TernaryConnect. These networks have emerged to facilitate hardware implementations of neural networks by reducing the memory requirements and removing multiplications. We applied our training method to BinaryConnect and TernaryConnect training algorithms: the obtained results are provided in Table 2. The source Python codes used for comparison are the same used in (Courbariaux et al. (2015); Lin et al. (2015)), available online (Lin et al. (2015)). The simulation results show that up to 70% and 80% of connections can be dropped by the proposed method from BinaryConnect and TernaryConnect networks without any compromise in performance without using data augmentation, respectively. Moreover, the binarized and ternarized sparsely-connected 50% improve the accuracy compared to the conventional binarized and ternarized fully-connected networks. Considering data augmentation (affine transformation), our method can drop up to 50% and 70% of connections from BinaryConnect and TernaryConnect networks without any compromise in performance, respectively. However, using data augmentation results in a better misclassification rate when it is used on networks trained with single-precision floating-point weights as shown in Table 2. In this case, our method still can drop up to 90% of connections without any performance degradation. It is worth specifying that we only

Table 2: Misclassification rate for a 784-1024-1024-1024-10 neural network on MNIST

| Method | Misclassification Rate (%) | | # of Parameters |
| | Without Data Augmentation | With Data Augmentation | |
|---|---|---|---|
| Single-Precision Floating-Point (SPFP) | 1.33 | 0.67 | 2913290 |
| Sparsely-Connected 50% + SPFP | 1.17 | 0.64 | 1458186 |
| Sparsely-Connected 90% + SPFP | 1.33 | 0.66 | 294103 |
| BinaryConnect[a] (Courbariaux et al. (2015)) | 1.23 | 0.76 | 2913290 |
| TernaryConnect[b] (Lin et al. (2015)) | 1.15 | 0.74 | 2913290 |
| Sparsely-Connected 50% + BinaryConnect[a] | 0.99 | 0.75 | 1458186 |
| Sparsely-Connected 60% + BinaryConnect[a] | 1.03 | 0.81 | 1167165 |
| Sparsely-Connected 70% + BinaryConnect[a] | 1.16 | 0.85 | 876144 |
| Sparsely-Connected 80% + BinaryConnect[a] | 1.32 | 1.06 | 585124 |
| Sparsely-Connected 90% + BinaryConnect[a] | 1.33 | 1.36 | 294103 |
| Sparsely-Connected 50% + TernaryConnect[b] | 0.95 | 0.63 | 1458186 |
| Sparsely-Connected 60% + TernaryConnect[b] | 1.05 | 0.64 | 1167165 |
| Sparsely-Connected 70% + TernaryConnect[b] | 1.01 | 0.73 | 876144 |
| Sparsely-Connected 80% + TernaryConnect[b] | 1.11 | 0.85 | 585124 |
| Sparsely-Connected 90% + TernaryConnect[b] | 1.41 | 1.05 | 294103 |

[a] Binarizing algorithm was only used in the learning phase and single-precision floating-point weights were used during the test run.
[b] Ternarizing algorithm was only used in the learning phase and single-precision floating-point weights were used during the test run.

used the binarized/ternarized algorithm during the learning phase, and we used single-precision floating-point weights during the test run in Section 4, similar to the approach used in (Lin et al. (2015)).

## 4.2 EXPERIMENTAL RESULTS ON CIFAR10

The CIFAR10 dataset consists of a total number of $60,000$ $32 \times 32$ RGB images. Similar to MNIST, we split the images into $40,000$, $10,000$ and $10,000$ training, validation and test datasets, respectively. As our model, we adopt a convolutional network comprising $\{128\text{-}128\text{-}256\text{-}256\text{-}512\text{-}512\}$ channels for six convolution/pooling layers and two 1024-node fully-connected layers followed by a classification layer. This architecture is inspired by VGGNet (Simonyan & Zisserman (2014)) and was also used in (Courbariaux et al. (2015)). Hinge loss is used for training with batch normalization and a batch size of 50.

In order to show the performance of the proposed technique, we use sparsely-connected networks instead of fully-connected networks in the convolutional network. Again, we compare our results with the binarized and ternarized models since they are the most hardware-friendly models reported to-date. As summarized in Table 3, simulation results show significant improvement in accuracy compared to the ordinary network while having significantly fewer parameters.

## 4.3 EXPERIMENTAL RESULTS ON SVHN

SVHN dataset contains $32 \times 32$ RGB images ($600,000$ images for training and roughly $26,000$ images for testing) of street house numbers. Also, $6,000$ images are separated from the training part for validation. Similar to the CIFAR10 case, we use a convolutional network comprising $\{128\text{-}128\text{-}256\text{-}256\text{-}512\text{-}512\}$ channels for six convolution/pooling layers and two 1024 fully-connected layers followed by a classification layer. Hinge loss is used as the cost function with batch normalization and batch size of 50.

Table 4 summarizes the accuracy performance of using the proposed sparsely-connected network in the convolutional network model, compared to the hardware-friendly binarized and ternarized

Table 3: Misclassification rate for a Convolutional Network on CIFAR10

| Method | Misclassification Rate (%) | | # of Parameters |
|---|---|---|---|
| | Without Data Augmentation | With Data Augmentation | |
| Single-Precision Floating-Point (SPFP) | 12.45 | 9.77 | 14025866 |
| Sparsely-Connected 90% + SPFP | 12.05 | 9.30 | 5523184 |
| BinaryConnect [a] (Courbariaux et al. (2015)) | 9.91 | 8.01 | 14025866 |
| TernaryConnect [b] (Lin et al. (2015)) | 9.32 | 7.83 | 14025866 |
| Sparsely-Connected 50% + BinaryConnect [a] | 8.95 | 7.27 | 9302154 |
| Sparsely-Connected 90% + BinaryConnect [a] | 8.05 | 6.92 | 5523184 |
| Sparsely-Connected 50% + TernaryConnect [b] | 8.45 | 7.13 | 9302154 |
| Sparsely-Connected 90% + TernaryConnect [b] | 7.88 | 6.99 | 5523184 |

[a] Binarizing algorithm was only used in the learning phase and single-precision floating-point weights were used during the test run.
[b] Ternarizing algorithm was only used in the learning phase and single-precision floating-point weights were used during the test run.

Table 4: Misclassification rate for a Convolutional Network on SVHN

| Method | Misclassification Rate (%) | Number of Parameters |
|---|---|---|
| Single-Precision Floating-Point | 4.734615 | 14025866 |
| BinaryConnect [a] (Courbariaux et al. (2015)) | 2.134615 | 14025866 |
| TernaryConnect [b] (Lin et al. (2015)) | 2.9 | 14025866 |
| Sparsely-Connected 90% + BinaryConnect [a] | 2.003846 | 5523184 |
| Sparsely-Connected 90% + TernaryConnect [b] | 1.957692 | 5523184 |

[a] Binarizing algorithm was only used in the learning phase and single-precision floating-point weights were used during the test run.
[b] Ternarizing algorithm was only used in the learning phase and single-precision floating-point weights were used during the test run.

models. Despite the fewer parameters that the proposed sparsely-connected network provides, it also yields state-of-the-art results in terms of accuracy performance.

## 4.4 COMPARISON WITH THE STATE OF THE ART

The proposed sparsely-connected network has been compared to other networks in literature in terms of misclassification rate in Table 5. In Section 4.1 to 4.3, we used the binarization/ternarization algorithm to train our models in the learning phase while using single-precision floating-point weights during the test run (i.e. inference phase). The first part of Table 5 applies the same technique, while in the second part we use binarized/ternarized weights also during the test run. We thus exploit a deterministic method introduced in (Courbariaux et al. (2015)) to perform the test run using binarized/ternarized weights. The weights are obtained as follows:

$$W_b = \begin{cases} 1 & \text{if } W \geq 0 \\ \text{-}1 & \text{otherwise} \end{cases},$$

$$W_t = \begin{cases} 1 & \text{if } W \geq \frac{1}{3} \\ 0 & \text{otherwise} \\ \text{-}1 & \text{if } W \leq \text{-}\frac{1}{3} \end{cases},$$

where $W_b$ and $W_t$ denote binarized and ternarized weights, respectively.

From the results presented in Table 5, we can see that our proposed work outperforms the state-of-the-art models with binarized/ternarized weights during the test run while achieving performance

Table 5: Misclassification rate comparison. Sparsity degree for the proposed network is $50\%$ in MNIST, and $90\%$ in SVHN and CIFAR10.

| Method | Datasets | | |
| | MNIST | SVHN | CIFAR10 |
|---|---|---|---|
| | Binarized/Ternarized Weights During Test Run | | |
| BNN (Torch7) (Courbariaux & Bengio (2016)) | 1.40% | 2.53% | 10.15% |
| BNN (Theano) (Courbariaux & Bengio (2016)) | 0.96% | 2.80% | 11.40% |
| (Baldassi et al. (2015)) | 1.35% | – | – |
| BinaryConnect (Courbariaux et al. (2015)) | 1.29% | 2.30% | 9.90% |
| EBP (Cheng et al. (2015)) | 2.2% | – | – |
| Bitwise DNNs (Kim & Smaragdis (2016)) | 1.33% | – | – |
| (Hwang & Sung (2014)) | 1.45% | – | – |
| **Sparsely-Connected + BinaryConnect** | **1.08%** | **2.053846%** | **8.66%** |
| **Sparsely-Connected + TernaryConnect** | **0.98%** | **1.992308%** | **8.24%** |
| Method | Single-Precision Floating-Point Weights During Test Run | | |
| TernaryConnect (Lin et al. (2015)) | 1.15% | 2.42% | 12.01% |
| Maxout Networks (Goodfellow et al. (2013)) | 0.94% | 2.47% | 11.68% |
| Network in Network (Lin et al. (2013)) | – | 2.35% | 10.41% |
| Gated pooling (Lee et al. (2015)) | – | 1.69% | 7.62% |
| **Sparsely-Connected + BinaryConnect** | **0.99%** | **2.003846%** | **8.05%** |
| **Sparsely-Connected + TernaryConnect** | **0.95%** | **1.957692%** | **7.88%** |

close to the state-of-the-art result of the model with no binarization/ternarization in the test run. The former are the most suitable and hardware-friendly models for hardware implementation of DNNs: our model shows a better performance in terms of both accuracy/misclassification rate and memory requirements. The obtained results suggest that the proposed network acts as a regularizer to prevent models from overfitting. Similar conclusions were also obtained in (Courbariaux et al. (2015)). It is worth noting that no data augmentation was used in our simulations throughout this paper except for the results reported in Table 2 and Table 3.

## 5 VLSI IMPLEMENTATION OF SPARSELY-CONNECTED NEURAL NETWORKS

In this Section, we propose an efficient hardware architecture for the proposed sparsely-connected network. In fully-connected networks, the main computational core is the matrix-vector multiplication that computes (1). This computation is usually implemented in parallel on GPUs. However, parallel implementation of this unit requires parallel access to memories and causes routing congestion, leading to large silicon area and power/energy consumption in customized hardware. Thus, VLSI architectures usually opt for semi-parallel implementations of such networks. In this approach, each neuron performs its computations serially, and a certain number of neurons are instantiated in parallel (Moreno et al. (2008)). Every neuron is implemented using multiply-and-accumulate (MAC) units as shown in Fig. 3(a). The number of inputs of each neuron determines the latency of this architecture. For example, considering a hidden layer with 1024 inputs and 1024 outputs, 1024 MACs are required in parallel and each MAC requires 1024 clock cycles to perform computations of this layer. In general, a counter is required to count from 0 to $N - 1$ where $N$ is the number of inputs of each neuron. It provides the addresses for the memory in which a column of the weight matrix $W$ is stored. In this way, each input and its corresponding weight are fed to the multiplier every clock cycle (see Fig. 3(a)). For binarized/ternarized networks, the multiplier in 3(a) is substituted with a multiplexer.

In Section 3, we described the formation of the Mask matrix $M$ using an SNG unit (see Fig. 2(a)). The value of $p$, through which it is possible to tune the sparsity degree of networks, also corresponds to the occurrence of 1 in a binary stream generated by SNG. Therefore, we can save up to $90\%$ of memory by storing only the weights corresponding to the 1s in the SNG stream. For instance,

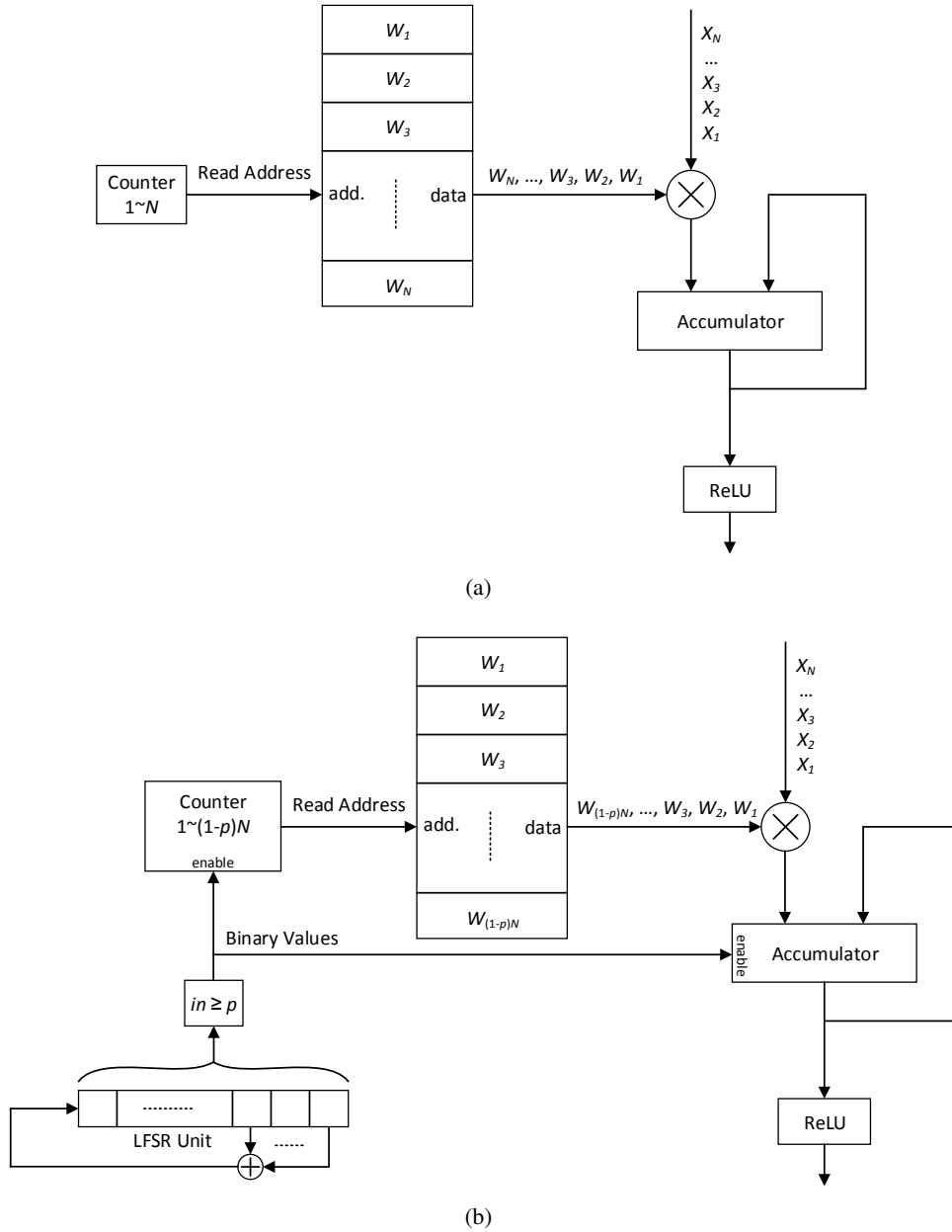

(a)

(b)

Figure 3: (a) shows the conventional architecture of a single neuron of a fully-connected network. (b) shows the proposed architecture of a single neuron of a sparsely-connected network.

considering a Mask matrix $M$ in Fig. 2(a), $W_s$ is formed as

$$
W_s = \begin{bmatrix}
0 & 0 \\
W_{21} & W_{22} \\
W_{31} & 0 \\
W_{41} & 0 \\
0 & W_{52} \\
W_{61} & W_{62} \\
0 & W_{72}
\end{bmatrix},
$$

Table 6: ASIC Implementation Results for a Single Neuron of Sparsely-Connected Network @ 400 MHz in TSMC 65 nm CMOS Technology.

| | Sparsity Degree | | | | |
|---|---|---|---|---|---|
| | $p = 0$ | $p = 0.5$ | $p = 0.75$ | $p = 0.875$ | $p = 0.9375$ |
| | Fully-Connected (FC) | Sparsely-Connected | Sparsely-Connected | Sparsely-Connected | Sparsely-Connected |
| Memory Size [bits] | 1024 | 512 | 256 | 128 | 64 |
| Area [$\mu m^2$] (improvement w.r.t. FC) | 26265 | 13859 (47% ↓) | 7316 (72% ↓) | 4221 (84% ↓) | 2662 (90% ↓) |
| Power [$\mu$W] | 278 | 155 | 86 | 60 | 43 |
| Energy [pJ] (improvement w.r.t. FC) | 712 | 397 (44% ↓) | 220 (69% ↓) | 154 (78% ↓) | 110 (84% ↓) |
| Latency [$\mu s$] | 2.56 | 2.56 | 2.56 | 2.56 | 2.56 |

and the compressed matrix $W_c$ stored in on-chip memories is

$$W_c = \begin{bmatrix} W_{21} & W_{22} \\ W_{31} & W_{52} \\ W_{41} & W_{62} \\ W_{61} & W_{72} \end{bmatrix}.$$

The smaller memory can significantly reduce the silicon area and the power consumption of DNNs architectures. Depending on the value of $p$, the size of the memory varies. In general, the depth of the weight memory in each neuron is $(1 - p) \times N$.

Fig. 3(b) depicts the architecture of a single neuron of the proposed sparsely-connected network. Decompression is performed using an SNG generating the enable signal of the counter and accumulator. Inputs are fed into each neuron sequentially in each clock cycle. If the output of the SNG is 1, the counter counts upward and provides an address for the memory. Then, the multiplication of an input and its corresponding weight is computed, the result stored in the internal register of the accumulator. If instead the output of the SNG is 0, the counter holds its previous value, while the internal register of the accumulator is not enabled, and does not load a new value. The latency of the proposed architecture is the same as that of the conventional architecture.

Table 6 shows the ASIC implementation results of the neuron in Fig. 3(b) supposing 1024 inputs. The proposed architectures were described in VHDL and synthesized in TSMC 65 nm CMOS technology with Cadence RTL compiler, for different sparsity degrees $p$. For the provided syntheses we used a binarized network. Implementation results show up to 84% decrement in energy consumption and up to 90% less area compared to the conventional fully-connected architecture.

## 6 CONCLUSION

DNNs are capable of solving complex tasks: their ability to do so depends on the number of neurons and their connections. Fully-connected layers in DNNs contain more than 96% of the total neural network parameters, pushing the designers to use off-chip memories which are band-width limited and consume large amounts of energy. In this paper, we proposed sparsely-connected networks and their training algorithm to substantially reduce the memory requirements of DNNs. The sparsity degree of the proposed network can be tuned by an SNG, which is implemented using an LFSR unit and a comparator. We used the proposed sparsely-connected network instead of fully-connected networks in a VGG-like network on three commonly used datasets: we achieved better accuracy results with up to 90% fewer connections than the state of the art. Moreover, our simulation results confirm that the proposed network can be used as a regularizer to prevent models from overfitting. Finally, we implemented a single neuron of the sparsely-connected network in in 65 nm CMOS technology for different sparsity degrees. The implementation results show that the proposed architecture can save up to 84% energy and 90% silicon area compared to the conventional fully-connected network while having a lower misclassification rate.

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
