# Peer review of "Sparsely-Connected Neural Networks: Towards Efficient VLSI Implementation of Deep Neural Networks"

_ICLR 2017 — accepted_

[Official Review · AnonReviewer3 · rating 6 · confidence 3 · 16 Dec 2016]
**Baseline results...**

From my original comments:

The results looks good but the baselines proposed are quite bad.

For instance in the table 2 "Misclassification rate for a 784-1024-1024-1024-10 " the result for the FC with floating point is 1.33%. Well far from what we can obtain from this topology, near to 0.8%. I would like to see "significant" compression levels on state of the art results or good baselines. I can get 0,6% with two FC hidden layers...

In CIFAR-10 experiments, i do not understand  why "Sparsely-Connected 90% + Single-Precision Floating-Point" is worse than "Sparsely-Connected 90% + BinaryConnect". So it is better to use binary than float. 

Again i think that in the experiments the authors are not using all the techniques that can be easily applied to float but not to binary (gaussian noise or other regularizations). Therefore under my point of view the comparison between float and binary is not fair. This is a critic also for the original papers about binary and ternary precision. 

In fact with this convolutional network, floating (standard) precision we can get lower that 9% of error rate. Again bad baselines.

----

The authors reply still does not convince me.

I still think that the same technique should be applied on more challenging scenarios.

[Public Comment · (anonymous) · 17 Dec 2016]
**related work**

A related work:

[Official Review · AnonReviewer1 · rating 6 · confidence 4 · 18 Dec 2016 (modified: 13 Jan 2017)]
**No Title**

The paper proposes a sparsely connected network and an efficient hardware architecture that can save up to 90% of memory compared to the conventional implementations of fully connected neural networks. 
The paper removes some of the connections in the fully connected layers and shows performance and computational efficiency increase in networks on three different datasets. It is also a good addition that the authors combine their method with binary and ternary connect studies and show further improvements.
The paper was hard for me to understand because of this misleading statement: In this paper, we propose sparsely-connected networks by reducing the number of connections of fully-connected networks using linear-feedback shift registers (LFSRs). It led me to think that LFSRs reduced the connections by keeping some of the information in the registers. However, LFSR is only used as a random binary generator. Any random generator could be used but LFSR is chosen for the convenience in VLSI implementation. 
This explanation would be clearer to me: In this paper, we propose sparsely-connected networks by randomly removing some of the connections in fully-connected networks. Random connection masks are generated by LFSR, which is also used in the VLSI implementation to disable the connections.
Algorithm 1 is basically training a network with back-propogation where each layer has a binary mask that disables some of the connections. This explanation can be added to the text.
Using random connections is not a new idea in CNNs. It was used between CNN layers in a 1998 paper by Yann LeCun and others:

[Official Review · AnonReviewer2 · rating 7 · confidence 3 · 20 Dec 2016]
**Neural Nets for embedded devices**

Experimental results look reasonable, validated on 3 tasks. 
References could be improved, for example I would rather see
Rumelhart's paper cited for back-propagation than the Deep Learning book.

[Final Decision · Program Chairs · 06 Feb 2017]
**ICLR committee final decision**

After discussion, the reviewers unanimously recommend accepting the paper.